# Piloting the Use of Concept Mapping to Engage Geographic Communities for Stress and Resilience Planning in Toronto, Ontario, Canada

**DOI:** 10.3390/ijerph182010977

**Published:** 2021-10-19

**Authors:** Martha Ta, Ketan Shankardass

**Affiliations:** 1Department of Psychology, Wilfrid Laurier University, Waterloo, ON N2L 3C5, Canada; taxx9300@mylaurier.ca; 2Department of Health Sciences, Wilfrid Laurier University, Waterloo, ON N2L 3C5, Canada; 3MAP Centre for Urban Health Solutions, Li Ka Shing Knowledge Institute, St. Michael’s Hospital, Toronto, ON M5B 1T8, Canada

**Keywords:** neighborhoods, chronic stress, community engagement, concept mapping, social media

## Abstract

The physical and social characteristics of urban neighborhoods engender unique stressors and assets, contributing to community-level variation in health over the lifecourse. Actors such as city planners and community organizations can help strengthen resilience in places where chronic stress is endemic, by learning about perceived stressors and assets from neighborhood users themselves (residents, workers, business owners). This study piloted a methodology to identify Toronto neighborhoods experiencing chronic stress and to engage them to identify neighborhood stressors, assets, and solutions. Crescent Town was identified as one neighborhood of interest based on relatively high levels of emotional stress in Twitter Tweets produced over two one-year periods (2013–2014 and 2017–2018) and triangulation using other neighborhood-level data. Using concept mapping, community members (*n* = 23) created a ten-cluster concept map describing neighborhood stressors and assets, and identified two potential strategies, a Crescent Town Residents’ Association and a community fair to promote neighborhood resources and build social networks. We discuss how this knowledge has circulated through the City of Toronto and community-level organizations to date, and lessons for improving this methodology.

## 1. Introduction

### 1.1. Chronic Stress and Community Resilience Interventions

Health inequalities are commonly measured within urban settings, with neighborhood-level variation in health outcomes routinely identified [1,2]. Efforts to study and conceptualize the causes of neighborhood variation typically focuses on indicators related to environmental hazards (e.g., air or water quality, environmental injustice), services (e.g., health care and other social services, public transportation, education), resources (e.g., greenspaces, arts/culture), or social composition (e.g., trust, political engagement, norms, inter-group relations) [3,4]. There is increasing recognition of the role of chronic stress as an important mediator of intra-urban inequalities [5]; however, action to identify and address the sources of chronic stress at the neighborhood level are uncommon.

Chronic stress is an overlooked target for health promotion and prevention in cities. Chronic stress can be described as a type of toxic stress (as opposed to good or tolerable stress), where distress is felt over a long duration either because of a deeply affecting negative experience (e.g., death of a loved one, an assault), or because of a persistent source of concern (e.g., unemployment, poverty) [6]. Although stress is often conceptualized as a psychological process or state rather than a health outcome, a wealth of evidence and theory connects chronic stress to a range of physiological and psychological processes that directly cause mental and physical health outcomes. Allostatic load has been proposed as a way to conceptualize the gradual wear and tear of chronic stress on these processes, while coping with chronic stress can also lead to the uptake of healthy or unhealthy behaviors (or habits) that alter risk for health outcomes over time. Moreover, the behaviors and health outcomes that result from chronic stress can, in turn, cause additional stress (e.g., the stress of managing type II diabetes to avoid additional morbidity). In total, this highlights the multi-factorial and iterative nature of chronic stress [7].

Resilience describes how individuals respond and cope to adverse events or experiences [8]. In the face of stressful events, individuals who develop resilience to stressors are able to “maintain relatively stable, healthy levels of psychological and physical functioning” [9] (p. 12). Though resilience can occur at the individual level, there is a need to consider a collective approach, such that we are able to understand what assets and resources a collective group, such as a community, will need to respond to adverse situations in a healthy way [8]. Community resilience is proposed as a process where a community uses their collective assets towards positive functioning following stressful events or experiences [10]. The development of community resilience has been observed through interventions targeting the reduction of violence in low-income areas [11], climate change [12], and empowering victims of bombings to become active agents within their community [13].

Critically, the aforementioned community resilience interventions focused on the use of community engagement and a collaborative partnership with communities to decide upon actions and goals to create the desired impacts on their social environments. These interventions either aimed to incorporate empowerment as a method or noted empowerment as one of the study’s effects. An empowering approach where interventions engage neighborhood users is important because these individuals may be vulnerable and could be further marginalized if they do not have some control in the process [14]. Neighborhood users hold important knowledge about the structural and contextual factors affecting their community, and thus, how to address issues at stake. Further, the process of facilitating a resilience intervention can lead to the identification of community strengths, or assets which can help to build resilience as a process [15], and strengthen the capacity of neighborhood users to choose healthy coping strategies.

### 1.2. Environmental Determinants of Stress in Neighborhoods

A common feature of most cities globally is the stratified nature of neighborhoods by levels of residents’ income. Segregation by income is a common feature of many cities, which creates structural barriers to health for local residents and workers. At a macro or city level, income inequality leads to a decline in investment in social and environmental conditions, which can have a larger impact on more vulnerable communities. Income inequality can also create less resilience and increase health risks for some neighborhoods by negatively impacting social bonds, decreasing social resources, reducing trust and civic participation, and increasing crime [16]. Thus, at the meso-level, neighborhoods characterized by lower income are less likely to have a range of material and economic resources, and may also feature less social cohesion and support. At a micro-level, income inequality creates an emphasis on social hierarchies, where those on the lower end experience the chronic stress of social comparisons [17]. People with lower incomes may experience more exceptional and mundane problems in their lives, and be accustomed to coping with stress in less healthy ways. These result in negative health outcomes and chronic diseases for people in some neighborhoods, including residents, employees, and business owners [7].

There is substantial evidence found demonstrating the link between neighborhoods and health outcomes independent of individual factors, such that characteristics of where individuals live, play, and work can increase the rate of risky coping behaviors, such as smoking behavior [18] and negative health outcomes, such as heightened mortality risk [19]. The physical features; availability of space to live, play, and work; availability of public or private support services; reputation; and socio-cultural features of a neighborhood are all determinants of health [20].

The environmental determinants are related to natural (i.e., naturally occurring), built (for human activity), and social (e.g., sociocultural factors) characteristics of a neighborhood which can serve as targets for resilience interventions [7]. Natural environmental determinants, such as exposure to green space, have been largely discussed as predicting reductions in perceived stress in individuals [21,22,23], as well as reducing risks of negative health outcomes, such as cardiovascular disease [24]. Further, green space has also been discussed to be inversely linked to health inequalities related to income deprivation [25].

Environmental determinants can influence the type and rate of stressors experienced, and the availability of resources to cope with stress, both of which can affect chronic disease outcomes through pathophysiology and coping behaviors related to chronic stress (as described above in more detail). Echoing the importance of social support (a feature of the social environment), one study found during the COVID-19 pandemic, neighborhood social support could lower the psychological distress individuals experienced despite the stressors associated with COVID-19, including caring for their physical health and financial losses [26]. Though environmental determinants can be categorized as natural, built, or social characteristics, they may also overlap within these categories. One study discussed that built characteristics of neighborhoods, such as porches and stoops in front of buildings, were related to the perception of available social support, and in turn affect experiences of psychological distress [27].

### 1.3. Concept Mapping for Community Knowledge

Using a health promotion approach, city planners (e.g., urban planners, neighborhood planners, health planners) and community actors can improve neighborhood health equity by modifying salient features of the local and city level environment to ameliorate sources of stress and strengthen community resilience. As we argue above, this action should be based on knowledge held by neighborhood users and built on existing assets where possible. One way that researchers can contribute to these processes is by accurately and reliably measuring community knowledge, which can empower community action and city action to impact change within their community, home, and work settings.

Gathering community knowledge about the collective concerns of neighborhood users and existing neighborhood assets is not a straight forward task because stress is a relational process where individuals perceive threats and challenges in their changing context in unique ways, resulting in stressors that affect individuals in different ways, with some not affected at all. Similarly, individuals may perceive neighborhood assets differently. Furthermore, as individuals can take part in shaping their environments, they should also be engaged and informed of the stressors they are experiencing, and how these stressors can affect their health, in order to reduce these stressors and their impact [7].

Community members can be engaged through various methods, including the use of focus groups and surveys. Concept mapping has been argued to be a step forward from collecting data using these traditional methods because of its elements of participant engagement in creating new knowledge and triangulating data through a mixed-methods approach [28]. This method also overcomes some of the limitations of focus groups, such as dominating voices and lack of participant engagement [29], as all participants take part in brainstorming and structuring the data independently [30].

Concept mapping is based on constructivist epistemology, as the creation of concept maps were considered new and evolving knowledge created by participants [31]. The concept map is generated by concept mapping software, and it conceptualizes a framework of all participants’ ideas into clusters and the interrelationships between ideas to support evaluation or planning activities [32]. The final concept map displays participants’ unique ideas and the interrelationships of ideas through the proximity and grouping of ideas, as participants are asked to group similar ideas together [33]. Applying research studies to real-world settings often requires balancing conflicting stakeholder perspectives on the implementation of changes, and using concept mapping is one approach to consider how different groups of stakeholders may perceive and be impacted by the implementation of change [28].

A study found that community engaged or participatory research studies identify concept mapping as a suitable tool because of its required power sharing process [34]. Participants can be involved in all stages of concept mapping, from deciding the focus of concept mapping to the interpretation of the final concept map. It is recommended to include participants in all stages of concept mapping to strengthen outcomes of research and to improve the sustainability of the research findings [34].

### 1.4. Pilot Testing in Crescent Town, Toronto

As one example of an urban setting with many diverse communities, the City of Toronto faces segregation by socioeconomic conditions and polarization of low- and high-income neighborhoods, as middle-income neighborhoods decline and shift to the category of low-income [35,36]. It is predicted that this trend in the reduction of middle-income neighborhoods will continue if policies are not changed or implemented to reduce the income inequality experienced within the city. An analysis of emotional stress expressions on social media over two one-year periods (2013–2014 and 2017–2018), as well as other publicly available administrative data about Toronto neighborhoods from Wellbeing Toronto and Urban HEART @Toronto indicating the presence of stressors, resilience, and stress-related disease outcomes, indicated that there may be relatively high levels of chronic stress in the Crescent Town neighborhood of Toronto [37,38]. The social media analysis was conducted on Twitter Tweets gathered between 2013–2014 and 2017–2018 that were geotagged to within one of the 140 neighborhoods in the City of Toronto, to identify locations of high emotional stress [39]. This was accomplished through sentiment analysis of tweets using the Stresscapes ontology to identify stress scores, followed by neighborhood aggregation of stress scores [39]. Triangulation across datasets was used to identify candidate neighborhoods with higher levels of stress on social media or presence of stressors, lower indicators of resilience, and higher rates of stress-related disease outcomes [39]. The objective of this study is to pilot the use of concept mapping to engage neighborhood users in Crescent Town to generate knowledge for neighborhood resilience planning. The aim is to identify chronic stressors and existing assets associated with their community using a method that could be reproduced as part of an urban resilience strategy.

## 2. Materials and Methods

Concept mapping methods were used to enable community members in Crescent Town to identify neighborhood stressors and assets, and to strategize solutions that address stressors by drawing on assets. In preparation for concept mapping, we intended to recruit five participants to form a community advisory board (CAB) using purposive sampling to recruit participants who could provide relevant perspectives and knowledge about the Crescent Town neighborhood. The objective for the CAB was to provide input about the recruitment of concept mapping participants, such as ideas about how to recruit participants in the local setting. Another objective was to gather feedback on the methods and plans for disseminating the study’s findings, such as whether there were any recommended ways to disseminate the findings without disadvantaging the neighborhood. Lastly, the CAB helped to further understand the data analysis that indicated a high potential for chronic stress in Crescent Town. Recruitment for the CAB was conducted in-person at local businesses and public spaces, as well as online in neighborhood groups.

Participants for concept mapping were recruited through purposive sampling and snowball sampling, to ensure that we retained participants that can provide relevant expertise and contribute to the understanding of the community’s stress and resilience. It is generally recommended that between 10 and 40 participants are recruited to ensure a substantial number of perspectives and facilitating discussion is possible [30]. We intended to recruit a sample of at least 22 participants who identified as residents, employees within the community, and business owners. Interested candidates completed a screening survey and provided either an email address or phone number to be contacted on. Participants consented to participating in two consecutive weekly sessions for concept mapping.

Concept mapping is performed through six steps, including preparation, generation, structuring, representation, interpretation, and utilization of findings [33]. Preparation was done by researchers by identifying clear goals of the study, and in turn, recruitment of participants who were purposively sought to provide relevant ideas for the goals of the study. At the initial session, participants were asked to independently brainstorm and record their responses to two focal prompts. These prompts were intended to be simplistic to easily help participants generate responses.

One issue in this community that causes chronic stress for you and/or others over time is ________________.One existing asset in this community that benefits you and/or others over time is _______________.

We then entered all responses into groupwisdom (Concept Systems Inc., Ithaca, NY, USA), a concept mapping software for the next stage. We synthesized all statements, including removing statements that represented similar ideas and editing statements to have a concise list of unique statements. As such, all responses were meant to represent different ideas and not contain more than one unique idea. Participants were then asked to independently group responses based on similarities. It was explained that how participants considered similarities of responses was dependent on themselves, and there was not one correct method of categorizing based on similarities. Participants were then asked to rate the importance of each statement on a Likert scale as their last task for the first session of concept mapping.

How would you rate the importance of these neighborhood issues and assets from 1 (weak) to 5 (strong)?

We then entered participants’ grouping and rating data into groupwisdom to be represented visually in a 2D point map. The multidimensional scaling (MDS) values created through the software were used to visualize the results in a cluster map through agglomerative cluster analysis and Ward’s method. We then discussed the number of clusters, groupings of ideas that would appropriately reflect each group’s unique idea, such that each cluster reflects all the statements in that group. Finally, each cluster was labeled appropriately with its idea, with the intention of discussing the cluster map with the participants and revising it based on the discussion with participants. In addition to the cluster map, a cluster rating map was also prepared for participants to review, which depicts the average rating of each cluster of statements [23] and helps to visualize the rating of each cluster based on participants’ responses to how important each stressor or asset is.

During the interpretation stage, participants returned to review the 2D point map, cluster map, and cluster rating map to discuss their understanding of the findings and provide feedback on how the maps could be refined to improve the representation of their ideas. This session was audio recorded for further qualitative analysis of the ongoing themes within the discussion. Participants were then asked to discuss how to make use of these findings, and to facilitate this discussion, we used the structured exercise 1-2-4-All. This allowed participants to independently consider what was the most stressful problem based on the most highly rated cluster of stressors, and the most useful solution based on the most highly rated cluster of assets. In groups of two, they discussed their ideas, and following this, in groups of four they collaboratively shared their ideas and presented this discussion to the larger group of participants. The entire group then discussed together what they agreed or disagreed on in terms of neighborhood stressors, and what a solution building on existing assets could be.

## 3. Results

In total, 23 participants participated in concept mapping from the Crescent Town community. Eighteen (78%) participants identified as neighborhood residents, while three (13%) participants identified as people who worked in the community. Six (26%) participants identified as volunteers within the community. Lastly, one (4%) participant identified as a student, and one (4%) identified as a community organization member. Twenty-three participants brainstormed 260 statements to two statement prompts that were then consolidated into 233 unique statements, and following this, 14 participants sorted statements, and 16 participants rated statements.

As depicted in Figure 1, as participants brainstormed statements to two different prompts, the sorting of this data also reflects the similarity of the two groups of responses in relation to stressors and assets. Ten different clusters were identified based on participants’ sorting data, with six groups of clusters identifying different themes of chronic stressors seen on the right of the cluster map, and four groups of clusters identifying different themes of neighborhood assets as seen on the left of the cluster map.

As seen in Table 1, each cluster’s importance rating is depicted based on the average rating data of participants, and Crescent Town appearance, maintenance, and infrastructure was identified as the most important stressor cluster, and accessibility and security as the most important asset cluster.

### 3.1. Clusters of Neighborhood Stressors

Cluster 1, “Barriers to Access and Social Isolation”, depicts the physical barriers affecting neighborhood users’ access to resources and services, as well as social problems in the neighborhood that lead to feelings of isolation. These two groups of statements may have also been sorted together because they can be interrelated, such that one resident may experience physical barriers around their neighborhood that affects their ability to participate and have social interactions.

Statements within Cluster 2, “Need for Health and Social Support Services and Businesses” suggests the lack of needed services and businesses, including support services for physical and mental health and affordable businesses in the neighborhood. This cluster also encompasses statements indicating a lack of available social supports, including a lack of a local tenants’ association, lack of clarity of resources, and lack of social participation in maintaining the neighborhood.

The next cluster (Cluster 3) refers to the depicted problems within the neighborhood and its high-rise buildings’ cleanliness, maintenance, and aging infrastructure. As such, the third cluster was titled “Crescent Town Appearance, Maintenance, and Infrastructure”. This cluster also describes statements related to the neighborhood’s empty business locations, cleanliness, and need for maintenance. Other statements refer to the need for pest control, maintenance of broken or overcrowding elevators, and the overall infrastructure breaking down.

Cluster 4, “Social Disorder and Safety”, portrays statements indicating a lack of order or control of the neighborhood that causes concerns for disorganization and safety. This encompasses statements identifying the issue of littering in the neighborhood, overcrowding, and residents slamming their doors within apartment buildings. Some statements also point to neighborhood users’ safety concerns with aging infrastructure and a lack of necessary emergency devices.

The fifth cluster identifies two groups of issues relating to residents who are new immigrants, including the problems they face as well as the problems others have regarding newcomers within the neighborhood. Thus, “Newcomers and Community Integration” refers to statements of problems that newcomers typically face in a new country, such as employment, housing, a language barrier, and learning a new culture that they also face after settling in Crescent Town. Other statements indicate problems related to newcomers settling in Crescent Town, the existence of ethnic conflict, racism, and discrimination.

The last cluster of neighborhood stressors (Cluster 6), “Income Insecurity” portrays problems directly or indirectly related to financial worries, including unemployment rates, childcare expenses, and living expenses. This cluster also includes statements regarding high crime rates and feelings of isolation, which may be a result of or related to income insecurity.

### 3.2. Clusters of Neighborhood Assets

The first cluster of statements relating to neighborhood assets (Cluster 7) depicts how neighborhood users felt a sense of belonging and a sense of cohesion in Crescent Town (“Sense of Belonging and Social Cohesion”). This includes statements that depict participants feeling like they belong within a local network, and others that identify local factors that facilitate this, including the multicultural composition of Crescent Town, friendliness amongst people in the community, and opportunities for belonging such as community fairs and festivals.

Cluster 8 (“Community and Business Services”) includes statements that identified how community services (such as community centers) and commercial services were assets of the neighborhood, as well as the variety of reasons that participants identified them to be assets. For example, some participants identified the convenience of having these services located within walking distance in the neighborhood, while others described the value in having a variety of these services to visit for different purposes.

Aside from the physical facilities within the neighborhood, participants also identified and grouped together ways that the setting of people’s housing in Crescent Town served as assets. Cluster 9, “Housing setting” included how the natural environment (including parks and trails) allows neighborhood users to participate in physical or social activities outside of their homes. It also included statements unrelated to the natural environment, such as beneficial aspects of housing conditions and rent control that helps keep housing costs manageable.

Cluster 10, the last group of statements, was titled “Accessibility and Security” to illustrate the neighborhood being an accessible area for neighborhood users to travel by foot or public transit within and outside of the area. This cluster also encompasses statements regarding the sense of security and safety that some participants identified, including enhanced security in some areas and low crime rates.

### 3.3. Interpretation of Clusters and Utilization of Cluster Map

After the cluster maps were produced, participants exchanged their interpretation of the maps and proposed changes to the maps to improve its depiction of their perspectives on neighborhood stressors and assets. This includes changes to cluster labels, and inclusion or exclusion of statements within each cluster. Although where the statement is located on the map is fixed, it can be regrouped into a different cluster that is close to its location. As an example, Cluster 7 was originally titled “Sense of Belonging and Social Cohesion” prior to participants’ reviewing and discussion and was reframed to “Available Support System”, to focus on the available support system for members of the community.

After determining that participants collectively agreed upon the changes to the concept map, participants also strategized within small groups on how to use any of the brainstormed assets to improve the problem with the highest importance rating. A facilitation technique called 1-2-4-All was used here [40], where participants first spent one minute focusing on the issue independently; then spent two minutes exchanging ideas and further discussing them with a partner; then spent four minutes building on the main ideas of the pairs in groups of four participants; and finally spent five minutes sharing ideas across groups with all present participants.

First, participants targeted the cluster, Crescent Town Appearance, Maintenance, and Infrastructure, as it had the highest cluster importance rating of 3.60 and identified one problem within this cluster for the exercise. After the completion of this exercise, participants shared their understanding of the problem and its contributing or related issues with the rest of the participants and collectively identified two distinct stressors within this cluster for the next strategizing exercise, including (1) increasing rent and (2) building maintenance and accessibility.

Participants completed the exercise, 1-2-4-All, a second time to focus on potential assets that can be built upon to address the two identified stressors. Participants collectively identified two potential solutions, including a community fair and a Residents’ Association for the two issues at hand. It was identified through the exercise that identified experiencing income insecurity and secure employment that matches their experience level and skillset contributed to the stressor of increasing rent. Although there were assets of available supports and resources within the neighborhood, there was notable difficulty in accessing them, as identified within the group discussion and the brainstormed statements. A community fair was proposed to promote available services and to facilitate an opportunity for neighborhood members to build a social network that can potentially lead to job opportunities. A Residents’ Association was determined as a potential solution to facilitate the opportunity for tenants and owners to take collective action towards issues related to building maintenance and accessibility. During the initial brainstorming process, it was identified within the statements that tension existed between various groups, including tenants and owners of building units, and collectively dealing with issues that affected both groups can potentially relieve this tension. It was also discussed that Crescent Town held many community spaces where a Residents’ Association would be able to have regular meetings, including the Crescent Town Club, community centers, and the local libraries.

The discussion with the participants was concluded with the understanding that these potential solutions were only the beginning of the discussion to address the neighborhood issues identified that contributed to chronic stress for neighborhood users and negative health outcomes as a result. The collective knowledge of the community that was captured through concept mapping would be valuable and meaningful to share with members of the local city councilor’s team and community level organizations for further discussion on how to utilize these results.

## 4. Discussion

To prevent intra-urban health inequalities, interventions are needed to identify and address sources of chronic stress within neighborhoods, ideally in a way that builds on existing community assets. This pilot study focused on enabling this process by engaging people who live or work in one Toronto neighborhood, Crescent Town, to identify sources of chronic stress that could be addressed through interventions; as well as existing neighborhood assets that could be integrated into interventions. The concept map produced with the participants is a framework for action for Crescent Town that conceptualizes sources of chronic stress, neighborhood assets, and their interrelationships. Six clusters of chronic stressors and four clusters of assets were identified with participants. To explore the utility of this approach with members of the Crescent Town community, we also used one exercise to illustrate how knowledge about sources of chronic stress and existing assets can inform ideas for solutions to the most prevalent chronic stressors that were identified. First, a community fair was proposed to help cope with rising rent by mobilizing social networks to share employment opportunities in the neighborhood. Second, a Residents’ Association was proposed to address long-standing building maintenance and accessibility issues by using collective action with landlords.

Concept mapping is often used for community engagement purposes due to its power sharing process with participants [34]. In this pilot study, participants contributed to data collection by brainstorming statements in response to the focal prompts, interpreting the collective statements as they discussed refining the clusters to better represent their perspective, and utilizing this knowledge as they discussed ways to make use of the findings. Although this method has been used to engage geographic communities to identify relevant interventions for other issues (e.g., to tailor implementation strategies [41], to increase physical activity interventions [42], and to improve the physical and mental health of students [43]), this study contributes to the growing literature of community engagement methods by piloting a concept mapping approach to prevent chronic stress for geographic communities. Further, although the relevance of an asset-based approach when using concept mapping for public health practice has previously been identified [33], it is uncommon for two focal prompts to be used in order to identify both public health problems, as well as existing assets in the community. In working through an exercise to formulate possible solutions to the sources of stress for Crescent Town, this approach clearly allowed community members to recognize existing forms of community resilience and enabled them to envision strengths-based interventions.

As discussed, this pilot study contributes to the existing literature of community engagement methods to address health disparities [44,45,46], by using concept mapping to identify both neighborhood sources of chronic stress as well as neighborhood assets to facilitate resilience planning. As an action oriented study, it aimed to take an upstream approach to the prevention of chronic stress within the neighborhood by identifying these problems, the importance of these problems, and engaging neighborhood users to plan for methods to reduce these sources of chronic stress. The results of this study will contribute to the community and City of Toronto’s understanding of the ongoing sources of stress within the neighborhood, and help to facilitate neighborhood planning with the consideration of existing strengths within the neighborhood. This work will also further contribute to future academic research studies of the Crescent Town community.

The validity of knowledge created by using concept mapping with members of the Crescent Town community is supported by other research studies conducted previously and contemporaneously in Crescent Town. The findings of this study were similar to results found in a previous study on Crescent Town commissioned by the Crescent Town Club [47], including the challenges that are identified in the community for the Crescent Town Club and other community service providers. A common theme identified within both studies is the high proportion of newcomers and need for supports for the specific challenges that newcomers face in settling, such as language barriers, obtaining employment, and housing. Other themes of concern similarly identified focused on the physical infrastructure, individual and family issues, and community development. The Crescent Town Club study chose focus groups and interviews to study the issues of Crescent Town, and when compared to the results of this study, concept mapping identified similar neighborhood challenges through different groups of perspectives, including residents, workers and business owners, and other neighborhood users. However, as discussed earlier, the use of concept mapping in this study also made it possible for participants to be part of the data collection, analysis, and shaping of the findings. Participants generated, sorted, and rated the data they provided, which can counteract some limitations of focus groups, such as dominating voices or lack of engagement [29]. Participants also shaped the results and discussed how the results, including the cluster map and cluster rating map can be used further. In contrast to other research methods, such as interviews and focus groups, in which participants’ involvement only occurs during data collection.

Similar data regarding the neighborhood were also collected for the Toronto Strong Neighborhoods Strategy (TSNS) 2020, as Crescent Town was identified as one of 31 Neighborhood Improvement Areas (NIAs) [48,49]. This study’s findings overlapped with issues identified in the TSNS, including identifying employment opportunities, anti-violence programs, and anti-racism programs which were top community priorities, strengthening the concerns that these issues have a need to be addressed. Further, this study found Crescent Town’s appearance, maintenance, and infrastructure as one of the top priorities of this group of participants, as it was the highest rated cluster that was not highlighted as a top priority through the TSNS. These differences may be due to the focus of the TSNS on identifying problems that the City of Toronto could help to address; whereas some stressors affecting communities are related to other sources, such as land owners and landlords. In this way, by not restricting the types of problems that were in the scope of our concept mapping focal prompt, we may have gathered a more holistic understanding of community problems. Comparison to other studies conducted contemporaneously in Crescent Town suggests that independent research with community members can play an important role in both validating and expanding knowledge identified about neighborhood problems’ impacts.

The use of concept mapping for this pilot study identified and visually represented two groups of data in a way that can easily engage members of the public, including community members, leaders, and advocates to build solutions together with service providers and the City of Toronto. While we explored the development of solutions with our participants, this was not a key focus of our pilot study. Rather, an executive summary of the study and its findings were shared with the local city councilor’s team as well as the TSNS team, and discussions focused on how the findings of this study could strengthen their existing strategies to improve the neighborhood as well as highlight any areas that were not yet recognized or have been overlooked but should be addressed. Future studies may consider intentionally planning a knowledge mobilization process, and engaging stakeholders beyond the community of focus a priori to improve uptake and application of community knowledge.

It is unknown how the findings of this study have been utilized by stakeholders to date. However, recent events in Toronto, and Crescent Town specifically, during the COVID-19 pandemic indicate the salience of the study findings. As in many other parts of the world, economic changes and policy responses during the pandemic have resulted in many tenants being unable to continue to pay rent on time, which has resulted in a steep rise in the number of attempted and completed evictions in Toronto since March 2020, including in Crescent Town [50,51]. In Crescent Town, one response to this community stressor has been the formation of the Crescent Town Tenants Union in Spring 2020, which has advocated for fair solutions with landlords in the neighborhood (e.g., forgiving or discounting unpaid rent rather than deferring payments). The formation of this union appears to be unrelated to the 2019 pilot study, and is more likely related to the phenomenon of other tenant unions that have been created in Toronto during the pandemic. Still, it is striking that in the face of this housing stressor, the Crescent Town community has mobilized a solution to their problems that reflects both pre-existing problems with landlords and a solution that mirrors what the concept mapping participants identified as a strengths-based approach to improve their power in engaging with landlords.

The diversity of the concept mapping participants may have been a limitation in this study. The majority of participants identified as neighborhood residents, 47% of participants identified as other types of neighborhood users. As such, it may be possible that the study’s findings identified a greater focus on stressors and assets more likely to be identified by residents rather than workers, volunteers, students, and other neighborhood users (e.g., people who come to Crescent Town for leisure activities).

As expected in a study aiming to pilot a research method not traditionally conducted for the research’s purposes, various challenges were encountered within the process. One of these challenges being that by employing two focal prompts, focusing on stressors and assets of the neighborhood, a higher than average amount of expected brainstormed statements were generated as a result. As such, it would be difficult for participants to rate and sort all of the generated statements within the scheduled timeframe. Participants were then instructed to rate and sort statements that belonged to either the group of stressors or assets, and then paired with another participant who rated and sorted the other group of data. This paired data was then entered into group concept mapping software. As such, further analyses of how different neighborhood roles rated and sorted the groups of statements was not conducted and there may be some variability in the findings if it were to be repeated. Future studies may consider the challenges encountered in attempting to gather knowledge about two groups of data through the concept mapping process.

## 5. Conclusions

In understanding that chronic stress is multifactorial and iterative, this study’s findings presented the neighborhood users’ identified neighborhood issues that cause them chronic stress, including some physical, social, and natural attributes of the neighborhood. This community knowledge created opportunities for different stakeholders to engage in neighborhood planning, including which areas need more attention to be paid by policymakers and service providers to understand where gaps need to be bridged between services and neighborhood users. However, this study also highlights the valuable assets that make this neighborhood unique for its users and that Crescent Town is a resilient neighborhood in Toronto because of its neighborhood users, community services, and accessibility and security. This study contributes to the growing academic literature on methods to facilitate neighborhood planning for the prevention of chronic stress within urban settings using the concept mapping method. In particular, while it is common for cities to engage local communities about needs, this approach uses a structured, participant-driven process to conceptualize the ideas of neighborhood users about the sources of neighborhood stress as well as local assets. Further, while our initial triangulation of administrative data and social media analysis identified Crescent Town as a site of interest in relation to chronic stress, it did not reveal sources of chronic stress or information useful to overcoming chronic stress. The knowledge generated from concept mapping speaks directly to what community members themselves identified as causing chronic stress and what assets in the community could serve to buffer these sources of stress. Concept mapping also helped to identify the relative importance of these ideas and how they are related to each other. This valuable community knowledge can be used to build upon the strengths of the neighborhood to reduce health inequalities across urban neighborhoods.

## Figures and Tables

**Figure 1 ijerph-18-10977-f001:**
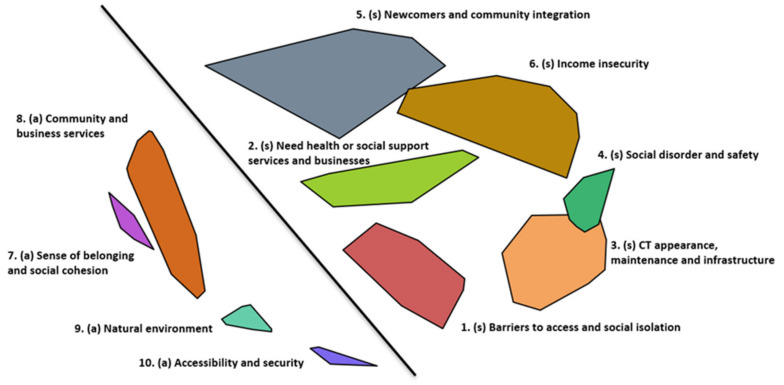
Cluster map of neighborhood chronic stressors (right) and assets (left) prior to stage of interpretation.

**Table 1 ijerph-18-10977-t001:** Importance rating of clusters of chronic stressors and assets.

Cluster	Chronic Stressors	Cluster Rating of Importance
1	Barriers to access and social isolation	2.91
2	Need health or social support services or businesses	3.30
3	Crescent Town appearance, maintenance, and infrastructure	3.60
4	Social disorder and safety	3.51
5	Newcomers and community integration	3.31
6	Areas in need of city support	3.27
**Cluster**	**Assets**	**Cluster Rating of Importance**
7	Available support system	3.67
8	Community and business services	3.78
9	Housing setting	4.16
10	Accessibility and security	4.48

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
