# Peer review of "Piloting the Use of Concept Mapping to Engage Geographic Communities for Stress and Resilience Planning in Toronto, Ontario, Canada"

_ijerph, 2021, doi:10.3390/ijerph182010977_

Round 1
Reviewer 1 Report
Dear authors.
Congratulations on the effort to develop the article.
I suggest some review points:
Page 2 and 3, Line 76 through 105: Include a more robust discussion using historical literature to date on Environmental Determinants of Stress in Neighborhoods;
Page 3, line 106 to 143: To further explore the discussion of the researched theoretical basis, I suggest adding textual elements on Concept mapping for community knowledge;
Page 4, line 159: Further develop the Theoretical Framework based on the consulted literature, discussing other elements that support the construction of this framework;
Page 4 and 5, line 159 to 220: Further detail the methodology construction process, explaining each step of the method until the analysis of the results;
Page 7, line 305: The text in the Neighborhood Asset Clusters section is unclear;
In Section 4 Discussion: The article fulfilled its purpose. However, I suggest expanding the analysis in order to identify the contributions of this research against the consulted literature;
Page 10, line 472: In the conclusions section, the authors perform a simple summary of the analysis of the results. I suggest highlighting the contributions to this thematic area, as well as the opportunities arising from this study.
I hope I contributed to the improvement of the study.
Reviewer 2 Report
This is a unusually polished piece of work, and presents in detail an interesting research approach to exploring community resilence/health. Overall, the approach is a pilot study, and thus the results are quite preliminary, and it would have been interesting to see a closing of the loop to see the ideas generated brought back to the community, and 'proofed' in practice. But it is a solid piece of work and I believe worthy of publication.
The target area for this study was chosen on the basis of social media and 'other' document analysis which is brushed over to a significant degree. I would have liked to see more detail here. Similarly the sampling method is quite loosely described.
Considering that the aim of the paper is to generate knowledge for future planning and identify chronic stressors, to some degree the social media analysis had already (it is argued) identified chronic stressors. It would be interesting to articulate how concept mapping would generate new knowledge relative to (for example) an analysis of Twitter data from the focus district. The paper articulates and demonstrates the advantages of the method over traditional focus group analysis, but does less of a thorough job of showing how the method improves on analysis of secondary data for example. I understand the approach is constructivist, but this begs the question, if the knowledge generators have been empowered by the process to understand and shape their environment, what next? The absence of a practice element, a bringing back to community, is in this sense, even more poignant.
The authors note that this is "the first study to pilot a concept mapping approach to prevent chronic stress for geographic communities". Statements like this area always a little risky, as the world of research is sprawling, and people can use different terminology to describe essentially the same thing. Risisky et al, for example, use concept mapping in Jackson, Mississippi to examine health disparities--using a different approach, but at the very least, worth mentioning as an example of this type of work in this broad field ("Concept Mapping as a Tool to Engage a Community in Health Disparity Intervention"). Note that I was able to find many studies in this journal alone using concept mapping.
Overall, however, a very solid paper.
Round 2
Reviewer 1 Report
I consider the improvements appropriate.